



# High Enrichment of Heavy Metals in Fine Particulate Matter through Dust Aerosol Generation

**Authors:** Qianqian Gao[1,2#], Shengqiang Zhu[1#], Kaili Zhou[1,2], Jinghao Zhai[3],

Shaodong Chen[1,2], Qihuang Wang[1,2], Shurong Wang[1], Jin Han[1,2], Xiaohui Lu[1,2],

Hong Chen[1], Liwu Zhang[1,2], Lin Wang[1,2], Zimeng Wang[1,2], Xin Yang[3], Qi Ying[4],

Hongliang Zhang*[1], Jianmin Chen[1,2]* and Xiaofei Wang*[1,2]

*1Shanghai Key Laboratory of Atmospheric Particle Pollution and Prevention, Department of Environmental Science and Engineering, Fudan University, Shanghai 200433, China*

*2Shanghai Institute of Pollution Control and Ecological Security, Shanghai 200092, China*

*3School of Environmental Science and Engineering, Southern University of Science and Technology, Shenzhen 518055, China*

*4Zachry Department of Civil Engineering, Texas A&M University, College Station, TX 77843, USA*

#These authors contributed equally to this paper

*To whom correspondence should be addressed.

Correspondence to:

Xiaofei Wang: Email: xiaofeiwang@fudan.edu.cn Tel: +86-021-31242526

Jianmin Chen: Email: jmchen@fudan.edu.cn Tel: +86(021)3124-2298

Hongliang Zhang: Email: zhanghl@fudan.edu.cn Tel: +86- 021-31248978



## Abstract

Dust is a major source of atmospheric aerosols. Its chemical composition is often assumed to be
similar to the parent soil. However, this assumption has not been rigorously verified. Here, we
generated dust aerosols from soils to determine if there is particle size-dependent selectivity of
heavy metals in the dust generation. Mn, Cd, Pb and other heavy metals were found to be highly
enriched in fine ($PM_{2.5}$) dust aerosols, which can be up to ~6.5-fold. To calculate the contributions
of dust to atmospheric heavy metals, regional air quality models usually use the dust chemical
profiles from the US EPA's SPECIATE database, which does not capture the correct size-dependent
selectivity of heavy metals in dust aerosols. Our air quality modeling for China demonstrates that
the calculated contribution of fine dust aerosols to atmospheric heavy metals, as well as their cancer
risks, could have significant errors without using proper dust profiles.

## Graphical Abstract

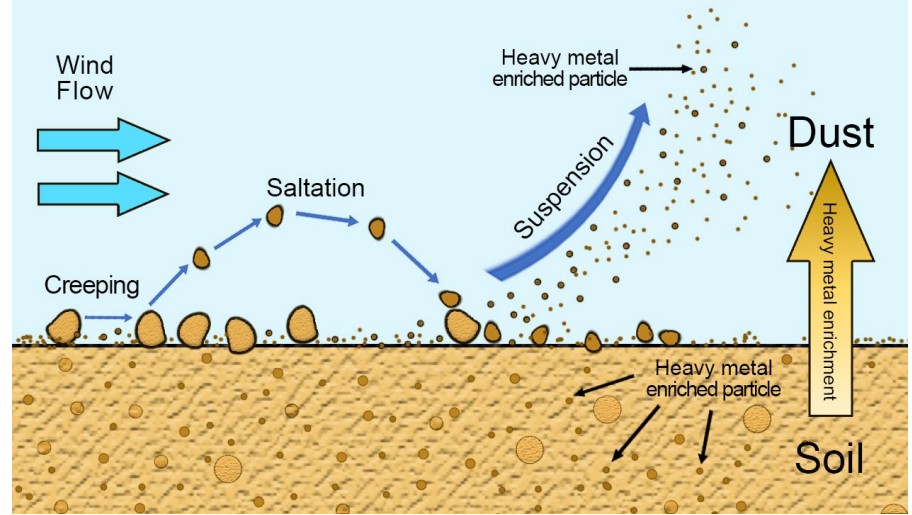







## Short Summary


Dust is a major source of atmospheric aerosols. Its chemical composition is often assumed to be
similar to the parent soil. However, this assumption has not been rigorously verified. Dust aerosols
are mainly generated by wind erosion, which may have some chemical selectivity. Mn, Cd and Pb
were found to be highly enriched in fine ($PM_{2.5}$) dust aerosols. In addition, estimation of heavy
metal emission from dust generation by air quality models may have errors without using proper
dust profiles.



# 1 Introduction

The major sources of natural aerosols include mineral dust aerosols produced by wind erosion (Prospero et al., 2002). Dust aerosols are influenced by regional atmospheric circulation, soil characteristics and local weather conditions (Bryant, 2013; Ding et al., 2005; Huebert et al., 2003; Liu et al., 2004; Yang et al., 2008), mainly generated and aerosolized when strong wind passes over soil or sandy areas (Gillette and Goodwin, 1974). Recent studies show mineral dust aerosol accounts for approximately 40% of the mass fraction of natural atmospheric aerosol, with an estimated annual flux of ~2,000 Tg·yr$^{-1}$ (Alfaro, 2008; Griggs and Noguer, 2002; Huneeus et al., 2011; Textor et al., 2006). As the second-largest natural source of atmospheric aerosols in terms of mass flux, dust aerosol has a profound impact on the ecosystem (Middleton et al., 2019), especially the climate (Evan et al., 2014; Kok et al., 2018; Shao et al., 2013). Interactions between dust aerosols and water vapor play a critical role in cloud condensation and ice nucleation processes (Kaufman et al., 2002; Tang et al., 2016). Dust particles can be transported on large scales (Shao and Dong, 2006), and could act as a medium to transport toxic compounds, including heavy metals, which significantly harm human health, particularly the human respiratory system and even cause premature death (Urrutia-Pereira et al., 2021).

Atmospheric studies often assume that the chemical composition of aerosolized dust is similar to the parent soil (Gunawardana et al., 2012; Zhuang et al., 2001). The chemical composition of dust aerosol consists of a key part in source apportionment modeling (Balakrishna and Pervez, 2009; Samiksha et al., 2017; Santos et al., 2017; Ying et al., 2018). A critical approach in source apportionment modeling is the chemical transport model, which predicts the dust aerosol on global



and regional scales based on the prior knowledge of source emission, atmospheric transport, and
chemical reaction process. SPECIATE is the EPA's speciation profiles repository of air pollution
sources of volatile organic compounds (VOCs) and particulate matter (PM). Therefore, the US
EPA's SPECIATE database is an important product to convert total emissions from specific sources
into the speciated emissions needed for the chemical transport model. The previous study has
combined the US EPA's SPECIATE database and air quality model to predict dust aerosols (Ying et
al., 2018), based on the assumption of the chemical composition of dust aerosols is similar to the
resuspended soil profiles.
Yet, dust generation and aerosolization are complex processes, which may have some chemical
selectivity. Most small dust particles (< 20 μm) are produced either by wind erosion, which leads to
soil movements such as creeping, saltation, and suspension (Burezq, 2020) or sandblasting process,
which leads soil particles (~75 μm) to be lifted by the wind, move in ballistic trajectories due to the
combined effect of aerodynamic force and gravity force (Grini and Zender, 2004; Shao and Raupach,
1993; Shao et al., 1996). The sandblasting efficiency of a soil particle is highly sensitive to its size
(Grini and Zender, 2004; Grini et al., 2002). In addition, the chemical composition of soil particles
can also vary with particle size. As smaller soil particles are more easily ejected, dust aerosol
particles are unlikely to have exactly the same composition as their parent soils (Perlwitz et al., 2015;
Wu et al., 2022). Dust deposited samples were the dust samples collected on the road or other
surfaces using a brush and plastic tray (Shangguan et al., 2022), while dust aerosol samples were
collected by filtering the air. Dust aerosols were produced by the ballistic impacts of wind-driven
sand grains (Kok et al., 2023). Indeed, some previous studies do find that in the deposited dust
samples (not dust aerosol samples), smaller particles tend to contain higher amounts of heavy metals



(Naderizadeh et al., 2016; Parajuli et al., 2016; Becagli et al., 2020). However, the heavy metal
profiles for dust aerosols from the US EPA's SPECIATE database seem to have no such enrichment
between each particle size, as Table S1 reports profile 41350 as an example. Although these profiles
have been widely used in air quality modeling works (Lowenthal et al., 2010; Simon et al., 2010;
Ashrafi et al., 2018), they were actually measured in the 1970s and 1980s with the resuspension of
soil samples, which placed soil in a glass tube and drew air flow to blow and suspend the soil
particles to the air (Miller et al., 1972). This method is not likely to produce realistic dust aerosols,
as it does not simulate sandblasting process properly. It is not known whether using such a
problematic dust profile could significantly impact air quality model calculations.

Here we examined the enrichment of heavy metals in the laboratory-generated dust aerosols. A dust
aerosol generator that mimics realistic sandblasting and saltation was used to generate dust aerosol
from a collection of soil samples (Lafon et al., 2014). The concentrations of heavy metals in soil
and dust aerosols were measured by an inductively coupled plasma mass spectrometer (ICP-MS).
In this study, some heavy metals, such as Mn, Cd, Zn and Pb, were found to be highly enriched in
dust aerosols. Especially, the enrichment factors would be much higher for smaller dust aerosols. In
addition, we also utilized a single particle aerosol mass spectrometer (SPAMS) to study heavy metal-
containing dust aerosols before, during, and after a dust storm. Regional air quality models usually
use problematic dust composition profiles from the US EPA's SPECIATE database. Herein we
modeled the contribution of dust aerosol to atmospheric heavy metal loadings, utilizing a range of
dust aerosol profiles determined in this laboratory study as well as the SPECIATE profile, to
investigate whether using a proper dust profile is critical to air quality modeling and cancer risk



calculations.

## 2 Materials and methods

### 2.1 Soil sample collection

Fourteen samples were collected from the top 10 cm of the natural soil profile from various locations
in dust source regions and Shanghai, China (Table S2, Fig. S1). S1-S4 were collected from dust
sources on the northern slope of Yinshan Mountain in central inner Mongolia and the adjacent areas
of the Hunshandake Sandy Land, S5-S12 were collected from dust sources of Hexi Corridor and
Alxa Plateau, S13 was collected in Xinjiang Province, in the dust sources of the Taklimakan Desert,
and S14 was sampled from Shanghai Yangpu District. As shown in Table S2, they represent several
soil types: S1 was silty loam; S2, S4, S7, S10, S11 and S12 were sand; S3 was sandy loam; S5 and
S6 were loam; S8 and S13 were loam sand; S9 and S14 were silty clay loam. Before dust aerosol
generation, soil samples were placed in a fume hood and left to dry, without stirring or other
treatment, before aerosolization. Fine and coarse dust aerosols ($PM_{2.5}$ and $PM_{10}$) were produced
with a GAMEL dust aerosol generator, which can realistically simulate the sandblasting process.
Then, the pH of the soil was measured. Detailed information can be found in Fig. S1 and Table S2.

### 2.2 Laboratory dust aerosol generation and collection

A laboratory dust generator (GAMEL: "Générateur d′Aérosol Minéral En Laboratoire") (Lafon et
al., 2014) was used to produce dust aerosols from the soil samples. The GAMEL dust generator can
realistically simulate the sandblasting process. In GAMEL's dust production system, 10 g of each
soil sample was added to a PTFE flask, which was agitated by a shaker simulating the sandblasting



process to produce dust aerosols. A constant flow of particle-free air was passed through the dust-
generating flask. The optimal generation parameter of the shaker was set at a frequency of 500
cycles/min according to Lafon et al., 2014 with an airflow rate of 8 liter/min controlled by a Mass
Flow Controller (MFC, Sevenstar, Beijing Sevenstar Flow Co., LTD). The sample stream was
filtered through a cyclone and particles were collected on a 47 mm PVC film held in a metal frame
filter holder (Pall Gelman, Port Washington, NY, USA). Dust-$PM_{2.5}$ and dust-$PM_{10}$ were obtained
with or without an 8LPM cyclone, respectively. The running time was 1min. To obtain more dust
aerosols in different size ranges, size-fractionated particle sampling of dust aerosols was carried out
with 10-stage Micro-Orifice Uniform Deposit Impactor (MOUDI 110R; MSP) with size cut points
of 10 μm, 5.6 μm, 3.2 μm, 1.8 μm, 1.0 μm, and 0.56 μm. Analysis of the size distribution and
chemical composition of dust generated by GAMEL and dust generated under natural conditions
has shown that the GAMEL generator can produce realistic dust aerosol (Lafon et al., 2014). All
the dust aerosol mass collected is shown in Table S3 and S4. The instrument setup is illustrated
in Fig. S2.

**2.3 Analysis of laboratory-generated dust aerosols**
The dust aerosol samples collected were weighed with an analytical balance and then put into 25 ml
digestion tubes with 6 ml 69 % $HNO_3$ symmetrically. The temperature program of Microwave
Digestion (Anton Paar) was as follows: initial temperature of 100 °C held for 5 min, then ramped
to 140 °C for 5 min, and finally at 180 °C for 60 min. The whole process was holding 120 min.
According to this study (Chang et al., 1984), almost all the heavy metal elements in the natural soil





and dust aerosol in concentrated nitric acid were extracted using this experimental procedure. After
digestion, the solution was acid-fed at 120 °C for 1.5 h, then deionized water (conductivity 18.25
MΩ) was added, the volume was constant with a 25 mL volumetric flask, and then passed through
a 0.45 μm membrane. The samples were diluted with 2 % $HNO_3$ by 4 times for further analysis.
Three blank PVC film samples were digested using the same method for background control.

The heavy metal content was determined by inductively coupled plasma mass spectrometer (ICP-
MS; Agilent, 8900). Before analysis, tuning procedures including plasma parameter, ion
transmission path, quadrupole mass spectrometer, and detector had been done. During analysis,
standard solutions were prepared at concentrations of 0, 1, 2, 5, 10, 20, 50, and 100 μg/L. "In, Bi,
and Rn" were used as internal standard elements, and were introduced into the nebulizer by mixing
with the sample to be tested and the standard solution in the sampling pipeline by online addition,
and the instrument drift and matrix effect were compensated. After each analysis of a sample, 2 %
dilute nitric acid was used to clean the injection line for 1 min, and then continue to collect the
second sample to eliminate the memory effect of the previous sample.
A scanning electron microscope (SEM; Phenom Pro) equipped with an energy-dispersive X-ray
detector was used for morphologies of particle examination at the voltage of 10 kV. All the samples
(soil, $PM_{2.5}$ and $PM_{10}$) were on the carbon conductive adhesive, then spray platinum to improve the
conductivity. Here, the parent soil of S10 and generated $PM_{2.5}$ and $PM_{10}$ were examined.
Statistical analysis was performed using SPSS Statistics. The correlation analysis was conducted
through Spearman's correlation and the significant difference was used with an independent sample
T-test.





**2.4 Ambient dust aerosol measurements**
On-site field measurements of ambient dust particles were conducted in Shanghai on May 23rd, 2018
(LT). The sampling was located on the sixth floor of the Environmental Science Building in
Jiangwan Campus, Fudan University, a typical residential area in a heavily polluted urban area. The
chemical composition of individual ambient particles was measured by single particle aerosol mass
spectrometry (SPAMS, Hexin Co., Ltd). Detailed information on SPAMS is available elsewhere (Li
et al., 2011). An adaptive resonance theory-based clustering method (ART-2a) was used to classify
the mass spectra generated and identify dust/heavy-metal-containing particles (Sullivan et al., 2007).
The Hybrid Single-Particle Lagrangian Integrated Trajectory HYSPLIT-4 model developed by the
ARL (Air Resources Laboratory) of the NOAA (National Oceanic and Atmospheric Administration),
USA, was employed to compute hourly resolved 48 h air mass backward trajectories at 500 m arrival
height (Lv et al., 2021; Pongkiatkul and Kim Oanh, 2007).

**2.5 Air quality model configuration and application**
The source-oriented CMAQ model v5.0.1 with an expanded SAPRC-99 photochemical mechanism
was applied to simulate $PM_{2.5}$ levels and track the sources of primary $PM_{2.5}$ ($PPM_{2.5}$) in China during
the entire year of 2013 (Guenther et al., 2012; Ying et al., 2018). The simulation domain covered
China and its surrounding countries, with a horizontal resolution of $36 \times 36$ km$^2$ ($127 \times 197$ grids).
Anthropogenic emissions were based on the Multi-resolution Emission Inventory for China (MEIC,
v1.3, $0.25° \times 0.25°$, http://www.meicmodel.org). Biogenic emissions were generated by the Model
of Emissions of Gases and Aerosols from Nature (MEGAN) v2.1 (Guenther et al., 2012). The





meteorological inputs for the CMAQ model were calculated by the Weather Research and
Forecasting (WRF) model (https://www2.mmm.ucar.edu/wrf/users).

Five major source contributions (windblown dust, residential, transportation, power generation and
industrial sources) to $PM_{2.5}$ were investigated based on the inventory-observation-constrained
emission factors (Ying et al., 2018). Three control trials were conducted for each heavy metal
according to the minimum, average, and maximum enrichment factors for the dust aerosols
generated from the soil collected from the four regions (three dust sources and Shanghai). A no-
enrichment trial was also conducted for comparison. It is noticeable that the enrichment factors
outside these four regions were estimated by inverse distance weight (IDW) spatial interpolation
methods (Zhang and Tripathi, 2018). The ratio of each heavy metal source contribution from dust
aerosol and all four sources were used to quantify the enrichment effect on heavy metal
concentrations in the atmospheric dust aerosols, which can be represented in Equation 1:
$$R = \frac{E_1 \times s_1 \times a}{\sum_{i=1}^{5} E_i \times s_i}$$
     Equation 1

Where $E_i$ is the $PPM_{2.5}$ emission from $i^{th}$ source, $s_i$ is the emission factor of the specific heavy metal
from $i^{th}$ source, $a$ is the enrichment factor of this heavy metal in dust-$PM_{2.5}$. *$E_1$, $s_1$, and $a$ are the*
*values for dust.*
In addition, the human health risk of heavy metals was assessed. Three main routes of chemical
daily intake (CDI, mg $kg^{-1}$ $day^{-1}$) of air heavy metals were: (1) direct ingestion of particles or gases
existed in the air (CDIing); (2) inhalation of suspended particles through mouth and nose (CDIinh);
and (3) daily absorption of heavy metals through skin (CDIdermal) (Luo et al., 2012). Specifically,
the carcinogenic and non-carcinogenic effects of heavy metals were assessed in the 13 age groups
in detail (from birth to $\leq$ 80 years old). CDIing, CDIinh, and CDIdermal were calculated as:





$$CDI_{ing} = C \times \frac{IRing \times EF \times ED}{BW \times AT} \times 10^{-6} \qquad \text{Equation 2}$$
$$CDI_{dermal} = C \times \frac{SA \times AF \times ABS_d \times EF \times ED}{BW \times AT} \times 10^{-6} \qquad \text{Equation 3}$$
$$CDI_{inh} = C \times \frac{IRinh \times ET \times EF \times ED}{BW \times AT} \times 10^{-6} \qquad \text{Equation 4}$$
Moreover, the total carcinogenic risk
(TCR) for each heavy metal were calculated by:
$$carcinogenic\ risk = CDI_{ing,dermal,inh} \times CSF \qquad \text{Equation 5}$$
$$TCR = \sum risk = CDI_{ing} \times CSF_{ing} + CDI_{inh} \times IUR +$$
$$CDI_{dermal} \times CSF_{ing}/ABS_{GI} \qquad \text{Equation 6}$$

Here the IRing was Ingestion rate (mg day$^{-1}$), EF was exposure frequency (day year$^{-1}$), ED was
exposure duration (year), BW was body weight (kg), AT was Averaging time (day), SA was total
body skin surface area (m$^2$), AF was skin adherence factor (mg cm$^{-2}$), ET was exposure time (hour
day$^{-1}$), ABSd was dermal absorption factor, IRinh inhalation rate (m$^3$ day$^{-1}$), ABS$_{GI}$ was
gastrointestinal absorption factor, CSF was cancer slope factor. The values of these parameters could
be found in the previous study (Gholizadeh et al., 2019).

## 3 Results and discussion


### 3.1 Enrichment of heavy metals in fine dust aerosols



. Fig. S3-S4 show the absolute concentrations of heavy metals in dust aerosols and their parent soils.
The concentrations of heavy metals in dust-PM$_{10}$ were similar to soil concentrations, which showed
a significant correlation between soils and PM$_{10}$ ($p<0.01$) (Fig. S5). While the concentrations of
heavy metals in dust-PM$_{2.5}$ were higher than those in soils, especially Mn, Ni, Cu and Zn, showed
significant differences ($p<0.001$) (Fig. S6). This trend was consistent across all soil samples. The





enrichment factor (EF) of heavy metals in dust aerosols relative to the parent soils was calculated
with Equation 8.
$$EF = \frac{C_1/m_1}{C_0/m_0} \text{.............................................................} \text{Equation 8}$$
Where $C_1$ is the heavy metal concentration in dust-PM; $m_1$ is the mass of dust-PM collected on the
filter; $m_0$ is the mass of soil in the ICP-MS sample, and $C_0$ is the heavy metal concentration of the
soil.

Figures 1 and S7 show that many heavy metals were highly enriched in fine dust aerosols (PM$_{2.5}$),
i.e., their absolute concentrations were significantly higher in fine dust particles than in the parent
soil (Fig. S6). V, Cr, Mn, Co, Ni, Cu, Zn, As, Cd, Ba, Ti, and Pb were all enriched in dust-PM$_{2.5}$
during the process of dust formation. The following trend of heavy metal enrichment was
established for dust-PM$_{2.5}$: Cd > Zn > Ba > Cu > Mn > Pb > Ni > Ti > Co > As > Cr > V. Notably,
the EFs of Cd were greater than 5 for soil S1, S10 and S11. Fig. 1 also illustrates that all heavy
metals were more highly enriched in smaller PM$_{2.5}$ dust particles compared to larger PM$_{10}$ dust
particles. For example, the Cd's EF reached ~6.4 and ~1.7 for dust-PM$_{2.5}$ and dust PM$_{10}$, respectively,
from soil S1. Most dust-PM$_{2.5}$ should originate from the small colloids in soil, which are defined as
soil particles with less than 2 μm in diameter. These soil colloids usually carry large amounts of
negative charges, which can help adsorb many cations in soil, including various heavy metal ions
(Brady and Weil, 2008). Thus, heavy metals are enriched in small soil aggregates. During the
sandblasting process, the smaller soil grains, with higher heavy metal concentrations, are more
likely to be ejected and form dust aerosols. The particle size dependence of heavy metal enrichment
could have significant ramifications for the health impacts of dust aerosols. The dust aerosol size



distribution of dust (Fig. S8) was also measured by an Aerodynamic Particle Sizer (APS,
APS Model 3321; TSI Inc.; USA). It is found that the peak of the particle size distribution of dust
was approximately at 2~3 μm.    Similarly, the scanning electron microscope (SEM) images of these
dust aerosols (generated by S10) also show the presence of a large number of particles with sizes of
2~3 μm. As particle size decreased, the shape of particles changed from flakes to rods, which means
a larger surface area (Fig. S9).    As for the influence of soil texture on dust aerosol enrichment, we
have not found any regularity and need to further explore.

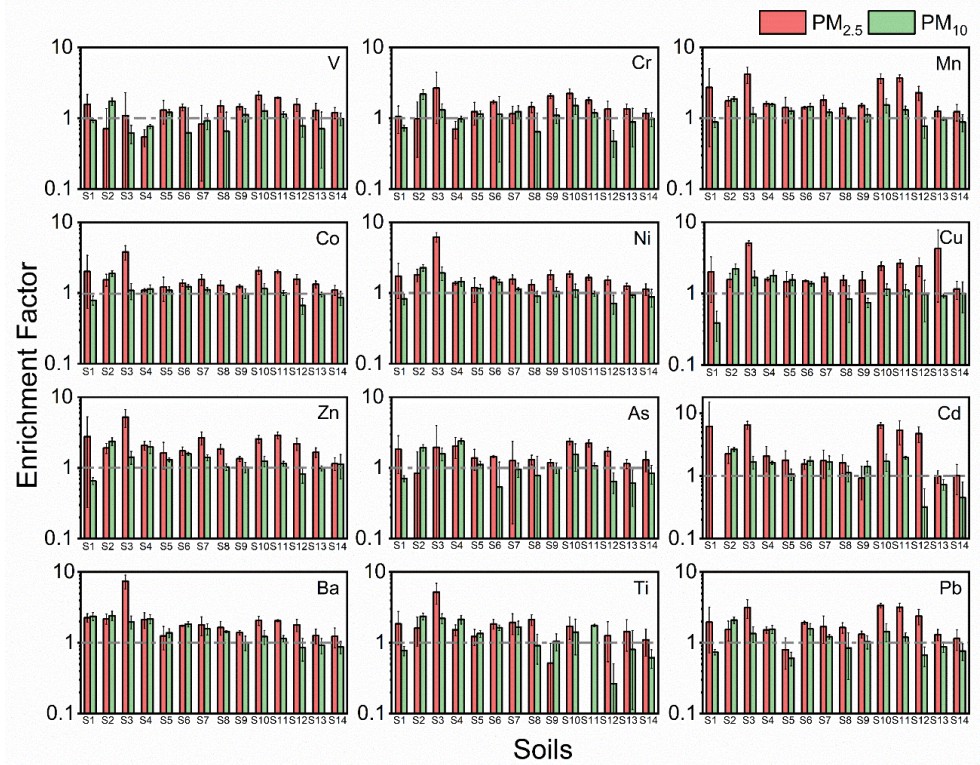


**Figure 1.** Enrichment Factors of $PM_{2.5}$ and $PM_{10}$. Enrichment factors of heavy metals in dust
aerosols from soil S1-S14; red represents $PM_{2.5}$ and green represents $PM_{10}$. The grey dotted line
represents the EF as 1. The whiskers on the bars represent the standard deviations of triplicates.

To investigate the link between dust particle size and heavy metal EFs in greater detail, a MOUDI
impactor was used to collect dust-PM from 0.56 to 10 μm (absolute concentration obtained in Fig.
S10). Consistent with the results discussed above, the EFs for some heavy metals, such as Pb,
significantly increased with decreasing particle diameter (r= -1, p < 0.01) (Fig. 2). For the smallest
dust particles (0.56~1.0 μm), the EFs for Pb was approximately 83, an order of magnitude greater
than the EFs (~3) for the largest dust particles (>10 μm). This result demonstrates that some heavy
metals are indeed enriched in smaller soil particles, which could be aerosolized during the
sandblasting process. The particle size dependence of heavy metal enrichment could have
significant ramifications for the health impacts of dust aerosols.

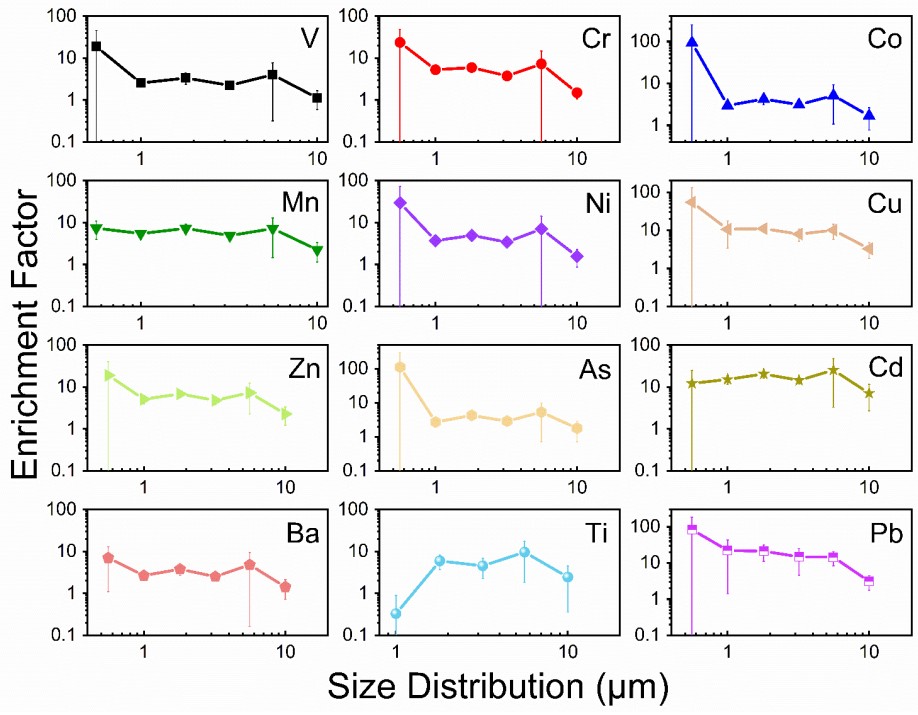


**Figure 2.** Enrichment factors of heavy metals in dust aerosols with different particle size ranges.
The EF data were produced from the Soil S10, with diameters at above10 μm, 5.6-10 μm, 3.2-5.6
μm, 1.8-3.2 μm, 1.0-1.8 μm and 0.56-1.0 μm. The whiskers on the bars represent the standard
deviations of triplicates.

**3.2 Modeling of the contributions of dust aerosols to atmospheric heavy metals using the dust profiles from this study and the SPECIATE datasets**

It is necessary to know the sources of atmospheric heavy metals to effectively control their emission.
Air quality models with emission inventories can estimate the contributions of various sources to
atmospheric heavy metals. However, when estimating heavy metal emissions from dust production,
some widely used air quality models, such as the CMAQ model, typically use dust profiles from the
US EPA's SPECIATE datasets. As discussed in the introduction, this dust profile may be outdated
and cannot reflect realistic dust compositions. We used the CMAQ model to assess the potential
impact of dust aerosol profile in atmospheric dust aerosol using our measured profile and the profile
(No. 41350) from the SPECIATE datasets. The model tracked heavy metals in $PM_{2.5}$ in China for
the year 2013 (see Methods) from five major sources: windblown dust, residential, transportation,
power generation, and industry.

Figure 3 shows the modeled contributions of the dust source to the Cr and Pb concentrations in
$PM_{2.5}$ for China, using the measured soil, dust-$PM_{2.5}$ profiles from this study, as well as the
SPECIATE composition profiles (see Methods). In addition, the modeled results for other metals,
such as As, Cr, Mn, Ti, and Zn were presented in Fig. S11-15.

For atmospheric Cr, it is clear that the scenario of applying SPECIATE database significantly
underestimates the contribution of dust aerosol, with the highest value of ~0.08    $\mu g/m^3$, when

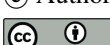



compared to the scenario of applying the measured dust-PM$_{2.5}$ profiles, which had the highest value
of ~0.14 $\mu g/m^3$. For Pb, as shown in the right column of Fig. 3, the scenario of applying
SPECIATE profile overestimates the contribution of dust aerosol, with the value up to ~0.4 $\mu g/m^3$,
when compared to the scenario of applying the measured dust-PM$_{2.5}$ profiles, which had the highest
value of ~0.14. These results demonstrate that the modeled heavy metal distribution in the
atmosphere is quite sensitive to the input of dust composition profile, strongly suggesting that using
a proper dust composition profile is a key in such air quality modeling.

As discussed in the Introduction, many atmospheric studies assume that dust aerosol composition
is similar to the composition of its parent soil. Here we also use the soil composition as an input
dust profile in the model calculation to see how the modeled results are compared to that using the
dust-PM$_{2.5}$ profile. For Cr, an obvious elevation of contribution was found by comparing the map
using soil (a) and dust-PM$_{2.5}$ (b) profiles, with the hotspots of contribution (~0.14 $\mu g/m^3$)
distributed in northwest China. The region with dust aerosol contribution ranged from 0.02 to 0.08
$\mu g/m^3$ covers most areas in China by using the dust-PM$_{2.5}$ profile. In contrast, the scenario of
applying soil profile shows smaller areas. For Pb, a significant difference is also found. The high
contribution areas are also mainly distributed in northwest China for scenarios of applying soil and
dust profiles, with the value up to 0.1 $\mu g/m^3$. While the area with low dust aerosol contribution
(<0.02 $\mu g/m^3$) shrinks considerably in the scenario of applying soil profile.

The applied dust enrichment factors to modeled Cr in PM$_{2.5}$ had an even stronger impact on modeled
source apportionment (Fig. 3a-3b). The average dust source contribution to the total PM$_{2.5}$ Cr



concentration over China was calculated to be 0.03, and 0.05 $\mu g/m^3$ in the scenarios of applying
soil and dust profiles, respectively. The model results for As, Cu, Mn, Ti and Zn (Fig. S11-S15) also
show similar trends, indicating applying realistic enrichment factors to heavy metal concentrations
in fine dust aerosols is critical to accurately model the sources of atmospheric heavy metals. These
results demonstrate that it is not appropriate to assume dust aerosol composition is equal to soil
composition, at least in air quality modeling.

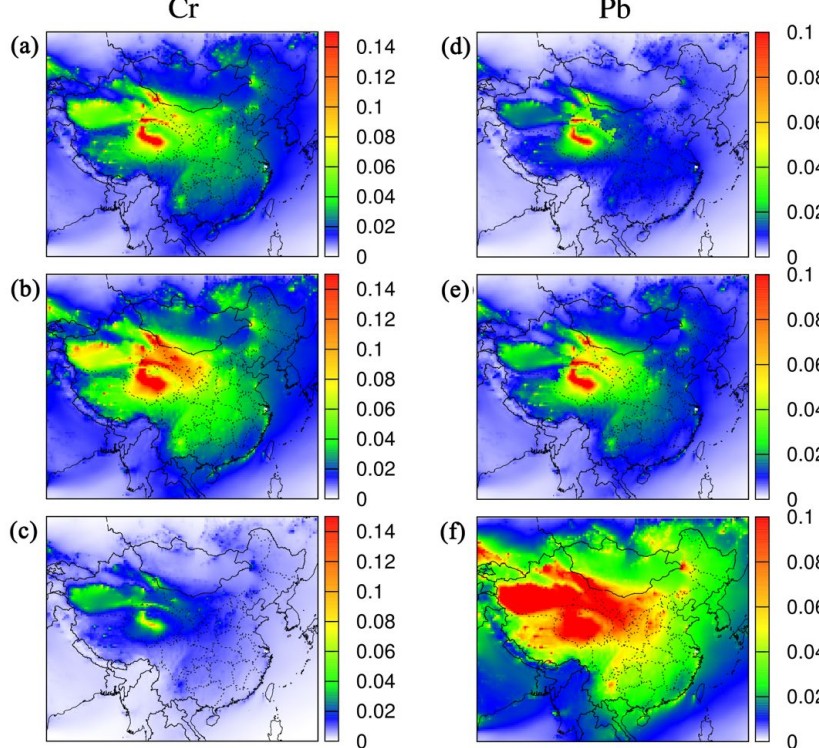


**Figure 3.** Modeling of the contributions of dust aerosols to atmospheric Cr and Pb concentrations.
These results use the dust profiles of measured soil (a, d), dust-PM$_{2.5}$ (b, e), and the SPECIATE
datasets (c, f). The unit is $\mu g/m^3$.
Figure 4 shows the total carcinogenic risk (TCR) of the modeled atmospheric heavy metals (Cu, Pb
and Zn) for each province in Mainland, China. The modeled results using the dust-PM$_{2.5}$ and the





SPECIATE profiles are compared here. The carcinogenic risks lower than $10^{-6}$ are considered
negligible, and risks above $10^{-4}$ are not accepted by most international regulatory agencies (Cheng
et al., 2015; Epa, 1989; Luo et al., 2012). For Cu, it is evident that using the SPECIATE profile
overestimated (the difference range up to ~ $7.5 \times 10^{-7}$) the TCR in China compared to using the dust-
PM$_{2.5}$ profile, as some regions exceed $10^{-6}$, the threshold value. For Pb, although all regions were
above $10^{-6}$, the TCR using the SPECIATE profile was greatly overestimated (the difference range is
~ $5.5 \times 10^{-6}$ - $4.0 \times 10^{-5}$). The model results for Zn showed that all regions were not above $10^{-6}$ but
significantly underestimated risks using the SPECIATE profile. This indicates that the health risk
assessment is also sensitive to dust composition profiles. Using the SPECIATE profile might be
problematic for assessing these risks.

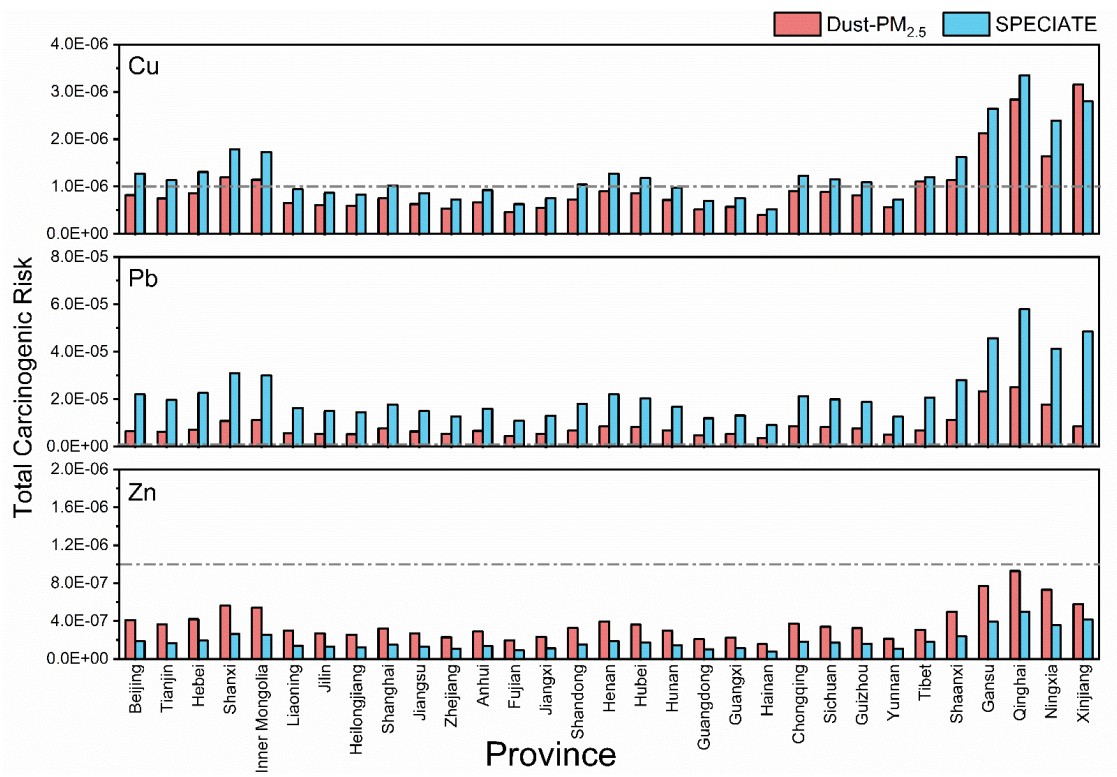




**Figure 4.** Comparison of the total carcinogenic risk (TCR) of the modeled atmospheric heavy metals

for each province in Mainland, China between using the dust-PM$_{2.5}$ and SPECIATE profiles. Here,

the TCR of Cu, Pb and Zn were calculated. The grey dotted line is 10$^{-6}$, the threshold value for

health concerns.

## 3.3 Field observation before, during and after a dust storm

Our modeling results demonstrate that dust aerosol is the main source of multiple heavy metals in

PM$_{2.5}$ in China. Therefore, dust storms should significantly increase the concentrations of heavy

metals in PM$_{2.5}$. To test this idea, we studied a dust-storm plume, which originated from Mongolia

and arrived in Shanghai (Huang et al., 2010) on 23 May 2018 (Fig. S16). Real-time single-particle

mass spectra were generated by a single-particle mass spectrometer. Single particle mass

spectrometry can offer detailed information on the chemically-resolved mixing state at the single-

particle level. According to the similarities of the mass-to-charge ratio and peak intensity of

characterized signals, *"Dust"* particles were classified via an adaptive resonance theory-based

clustering method (ART-2a, see Method). The number fraction of *Dust* particles was ~4.94% before

and after the dust storm and it increased to~9.73% during the dust storm episode (Fig. 5a).

*Dust* particle mass spectra also contained ion markers indicative of an array of heavy metals (*m/z*

55[Mn$^+$], 51[V$^+$], 207[Pb$^+$], 63[Cu$^+$], 75[As$^+$], 91[AsO$^+$], 52[Cr$^+$], -84[CrO$_2^-$], -100[CrO$_3^-$]) (red

sticks in Fig. S17), indicating the existence of heavy metals in the ambient dust aerosols. The time

series of Pb-containing and Cr-containing particle number fractions showed similar trends to the

*Dust* particles. When the dust storm arrived, both Pb-containing and Cr-containing particle fractions





increased as the dust cluster fraction increased. Before and after the dust storm, the percentages of
Pb-containing and Cr-containing particles that overlapped with the *Dust* cluster were 41% and 32%,
respectively. However, this overlapped ratio increased to 86% and 71% during the dust storm
episode. The increase of heavy metal particles in step with the dust particles indicated the dust
particles could be the dominant source of these heavy metal species during this dust storm episode.

We further analyzed the size-resolved number fraction of dust aerosol, Pb-containing, and Cr-
containing particles during the dust storm episode (Fig. 5b). The number fraction of *Dust* particles
increased with increasing aerodynamic diameter. For particles above 1.0 μm, *Dust* accounted
for >12% of the total particles during the storm. However, the Pb-containing and Cr-containing
particles made up a larger number fraction of analyzed particles with decreasing particle diameter
size (< 1 μm). The number fractions of Pb-containing and Cr-containing particles were 5.7% and
7.9% of all mass spectra for particles from 0.2-0.3 μm. This result was consistent with our laboratory
results that there is high heavy metal enrichment in smaller dust particles and our modeling results
that dust aerosol is likely a major source of atmospheric Pb and Cr over China.



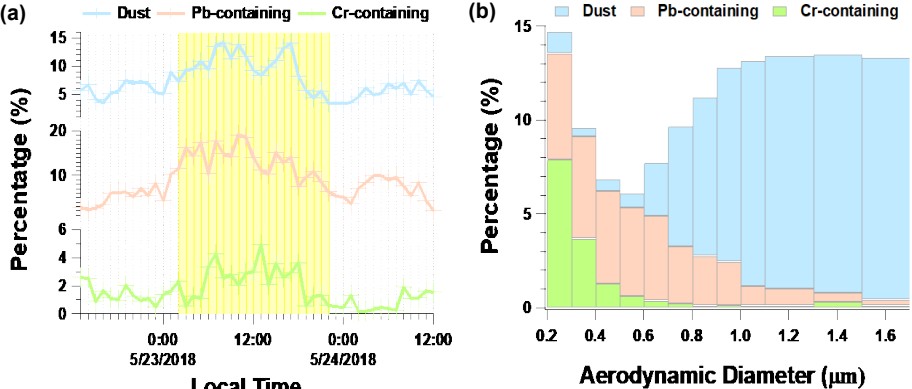


**Figure 5.** Ambient dust aerosol measurements. (a) Temporal variation of the percentages of dust
aerosol, Pb-containing, and Cr-containing particle clusters. The yellow shadow represents the dust
storm episode. (b) Size-resolved number fraction of dust aerosol, Pb-containing, and Cr-containing
particle clusters.

## 4 Environmental implications

In this study, many heavy metals were found to be highly enriched in fine ($PM_{2.5}$) dust aerosols
compared to their concentrations in the parent soils. We propose that heavy metals tend to be
enriched in smaller soil aggregates (Ikegami et al., 2014). During the sandblasting process, the
heavy metal enriched smaller soil aggregates are more likely to be ejected and form dust aerosols.
This work finds that dust aerosols from different soils may have a range of heavy metal enrichment
factors. To study the transfer of heavy metals from soils to the air, it is critical to have a complete
set of enrichment factors for each major soil type. There exists a difference among the heavy metal
enrichment factors from different soil samples. The variability in the EFs is likely due to differences
in soil properties (soil texture and size distribution etc.) which may affect the sandblasting/saltation





process. For example, the enrichment factors of heaviest metals for Soil S1, S10 and S11 were
higher than other soils. The detailed reason is still unknown and needs further exploration. Moreover,
air quality models, including CMAQ models and various CMB models, often use the dust chemical
profiles from the US EPA's SPECIATE to calculate the contribution of fine dust aerosols to
atmospheric heavy metals, which are outdated and could lead to significant errors in estimating the
emission of heavy metals through dust generation. Without using proper dust profiles in estimating
heavy metal emissions from dust generation, the contribution of fine dust aerosols to atmospheric
heavy metals, and its associated health risks are likely significantly mistaken.

## 5.Conclusions

Dust generation and aerosolization are complex processes that may have certain chemical selectivity.
Here, we deployed a laboratory generator to produce dust aerosol with a realistic sandblasting
process. The concentrations of heavy metals (including V, Cr, Mn, Co, Ni, Cu, Zn, As, Cd, Ba,
Ti, and Pb) in soils and fine ($PM_{2.5}$) and coarse ($PM_{10}$) dust aerosols were measured. With
research efforts to elucidate the enrichment process of heavy metal in dust aerosols comparing
to their parent soils, our results fill the knowledge gaps of the compositional variation of heavy
metal between the parent soils and the generated dust aerosols. Mn, Cd, Pb and other heavy
metals were found to be highly enriched in fine ($PM_{2.5}$) dust aerosols, which can be up to ~6.5-
fold. These findings were also consistent with our field observation results. In addition, air
quality models often use an outdated heavy metal profile for dust aerosols from the US EPA's
SPECIATE database, which seems to be lack of enrichment between each particle size. We modeled
the impact of the contribution of heavy metals in dust aerosol and their health risks in CMAQ,





436 a widely used air quality model, and determined that atmospheric heavy metal concentrations

437 over China, which drastically changed when we applied different dust profiles, such as the

438 measured soil, dust-$PM_{2.5}$ profiles from this study, as well as the SPECIATE composition

439 profiles. Our air quality modeling for China demonstrates that the calculated contribution of fine

440 dust aerosols to atmospheric heavy metals, as well as their cancer risks, could have significant errors

441 without using proper dust profiles.

## Supplement

443 The supplement related to this article is available online at: http://dx.doi.org/ 0.17632/byg6xk2fg9.1.

## Data availability

445 All data supporting this study and its findings will be available in an online data repository at:

446 http://dx.doi.org/10.17632/wpphf8rd33.1.

## Author contributions

448 X.W. and J.C. conceptualized the work and designed the experiments. H.Z. and S.Z. led the air

449 quality modeling work. Q.G. lead the experimental work of heavy metal enrichment measurements.

450 J.Z. led the field observation. K.Z., Q.W., S.C., S.W., J.H., X.L. and H.C. helped in experimental

451 works. L.Z., L.W., Z.W., X.Y. and H.Z. helped in the experimental design and data analysis. Q.Y.

452 provided the data required for the air quality modeling. All authors contributed to the paper's writing.

453



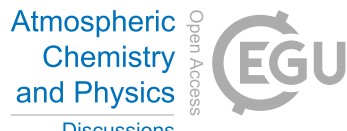

## Competing interests

The authors declare no competing interests.

## Disclaimer

Publisher's note: Copernicus Publications remains neutral with regard to jurisdictional claims in published maps and institutional affiliations.

## Acknowledgements

The authors also thank Xingxing Wang and Xiangcheng Zeng for their help in heavy metal measurement.

## Financial support

This work was partially supported by the National Natural Science Foundation of China (Nos. 92044301, 42077193, 21906024). Comments from Dr. Camille Sultana greatly improved this manuscript.

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
