# Peer review of "High Enrichment of Heavy Metals in Fine Particulate Matter through Dust Aerosol Generation"

_Atmospheric Chemistry and Physics, 2022_

## Author Comment (AC1)

**Responses to interactive comments**

**Journal: Atmospheric Chemistry and Physics**

**Manuscript ID: acp-2022-802**

**Title: High Enrichment of Heavy Metals in Fine Particulate Matter through Dust Aerosol Generation**

We appreciate Referee #1's comments and suggestions to help improve the manuscript. Every comment is addressed, and the detailed responses and related changes are shown below. Our response is in blue and the modifications in the manuscript are in red.

The manuscript entitled "High Enrichment of Heavy Metals in Fine Particulate Matter through Dust Aerosol Generation" examines the relationship between mineral dust aerosols and their parent soils in terms of heavy metal contents, demonstrating enrichment of heavy metals in the fine aerosol particles. The study combines laboratory and numerical modeling experiments and highlights that the heavy metal content may be higher than usually estimated by most models dealing with dust aerosols. Such findings are of great importance for air quality assessment and health implications as it evaluates the sensitivity of air quality models to the size-resolved heavy metal composition established through experimentation. The results are novel and significant and justify the publication of this paper. Nonetheless, I did notice some points that need to be clarified before the manuscript can move forward. Therefore, minor revision has to be done before this manuscript could be accepted for publication in the ACP.

**Response:** We thank Referee 1 for Referee #1's valuable comments and suggestions. Below are the responses to each specific comment.

**Major comments**

1. It seems that soil types play a quite important role in affecting the heavy metal

enrichment in dust aerosols. However, it is not clear how soil type was determined. Thus, the authors need to provide detailed information on how the soil type was determined. Please show the details in the Materials and Methods section.

**Response:**

We have included additional details on how the soil texture was determined in both the revised Materials and Methods section and the Supplementary Information (SI) section.

**Changes in manuscript:**

"Soil texture characterization was conducted based on the method outlined in a previous study (Kettler et al., 2001). Soil particle dispersion was achieved by adding hexametaphosphate (HMP) and sodium hydroxide (NaOH) to a soil sample (particle size< 2 mm) and shaking it for 16 hours. The percentage of sand and silt was obtained using a Laser Scattering Particle Size Distribution Analyzer (LA-960). Further details can be found in the SI."

**Texture S1.** Soil texture characterization

To measure the particle size distribution of the soil, approximately 0.03 to 0.5 g of air-dried soil samples were first passed through a 2 mm sieve. Subsequently, 10 mL of distilled water was added to the soil, and a dispersant was used to adjust the pH based on the soil's alkalinity or acidity. The dispersant consisted of either 1 to 1.5 mL of 0.5 mol/L hexametaphosphate (HMP) or 0.5 mol/L sodium hydroxide (NaOH). The mixture was then left to soak overnight before undergoing ultrasonic vibration for 2 minutes. Finally, the Laser Scattering Particle Size Distribution Analyzer (LA-960) was utilized to measure the soil samples labeled as S1-S14.

2. This paper used GAMEL system for laboratory simulation, which is a small scale dust generator. Please compared and explain the similarity and difference of the GAMEL system and the wind tunnel system.

**Response:**

The GAMEL system and wind tunnel system share the similarity of being able to realistically simulate dust aerosol generation. However, they differ in several

aspects:

Firstly, the wind tunnel typically requires tens of kilograms of soil, which makes it challenging to collect sufficient samples from field sites for conducting tests under various experimental conditions. In contrast, the GAMEL system only requires 10 grams of soil sample.

Secondly, the wind tunnel faces difficulties in preventing contamination from room air due to its large air flow rate. Filtration of such a high air flow would incur significant costs. On the other hand, the GAMEL system operates with an air flow rate of system is 8 LPM (liters per minute), allowing it to be equipped with a small HEPA filter that effectively removes ambient aerosols.

Based on the above considerations, we referenced this literature (Lafon et al., 2014) and used this GAMEL system to simulate the generation of dust aerosols.

**Changes in manuscript:**

"Wind tunnels have the advantage of realistically simulating the generation of dust aerosols. However, conducting this study has certain drawbacks. These include the requirement for a substantial quantity of parent soils and the significant cost associated with eliminating ambient aerosol interference (Alfaro et al., 1997; Lafon et al., 2006; Alfaro, 2008)."

3. How is total carcinogenic risk (TCR) calculated is not very clear. For example, what are the 13 age groups. Please provide more details of the calculation.

   **Response:**

   We have included additional details on the calculation of TCR (Target Cancer Risk) and referenced the following literature for this purpose (Gholizadeh et al., 2019; Luo et al., 2012; Moya et al., 2011; Doe, 2011).

   The 13 age groups are divided as follows (in years): <1, 1 to <2, 2 to <3, 3 to <6, 6 to <11, 11 to <16, 16 to <20, 21 to <31, 31 to <51, 51 to <61, 61 to <71, 71 to <81, and ≥81. The variables and values used for estimating human exposure to heavy metals were obtained from the U.S. Environmental Protection Agency (USEPA) and the U.S. Department of Energy (USDoE) (Moya et al., 2011; Doe,

2011) .

**Changes in Manuscript:**

"To assess the carcinogenic and non-carcinogenic effects of heavy metals, we evaluated these effects in 13 age groups ranging from birth to ≤80 years old. These age groups are as follows: <1, 1 to <2, 2 to <3, 3 to <6, 6 to <11, 11 to <16, 16 to <20, 21 to <31, 31 to <51, 51 to <61, 61 to <71, 71 to <81, and ≥81 years (Gholizadeh et al., 2019). The variables and values used for estimating human exposure to heavy metals were obtained from the U.S. Environmental Protection Agency (USEPA) and the U.S. Department of Energy (USDoE) (Moya et al., 2011; Doe, 2011)."

**Minor comments**

1. Line 137-139: In the GAMEL experiment, the author used an 8 L/min dust aerosol stream to sweep 10 g soil, can it generate dust aerosols stably? Is the particle size distribution related to the wind speed? Please explain.

   **Response:**

   The GAMEL generation system has the capability to stably generate dust aerosols. In our study, the experimental duration was set to 1 minute, and the particle size distribution was determined using the Aerodynamic Particle Sizer (APS). In this experiment, the graph below illustrates the observed particle size distribution, with a peak occurring at 2-3 μm. Although this experiment only investigated the particle size distribution generated by S9 and S14, one of our previous studies conducted with the same experimental setup exhibited a similar shape in particle size distributions (Gao et al., 2023). Hence, the GAMEL system can stably generate dust aerosols.

[Figure]

**This study:** Supplementary Figure S8. Particle size distribution of dust aerosols produced from soil S9 and S14. The size distribution was detected by an Aerodynamic Particle Sizer (APS), which size range is 0.5-20 μm.

[Figure]

**Previous study: Enrichment and Transfer of Polycyclic Aromatic Hydrocarbons (PAHs) through Dust Aerosol Generation from Soil to the Air (Gao et al., 2023):** The particle size distribution (PSD) of dust particles. All the soil were collected from the top 20 cm of the soil profile in different regions of Hunan province, south China. (a) is Besides the Pedestrian Road, (b) is Under the Street Pipe, (c) is Inside Baling Petrochemical Plant campus 1, and (d) is Inside Baling Petrochemical Plant campus 2.

In terms of the relationship between particle size distribution and wind speed, a study by (Kok, 2011) suggests that the size distribution of dust aerosols released naturally remains unchanged, regardless of any changes in wind speed.

2.  Line 153: Please remove the space between "69" and "%".

    **Response:**

    Revised accordingly.

    **Changes in Manuscript:**

    "…69% $HNO_3$…".

3. Line 160 and Line 169: Please remove the space between "2" and "%".

**Response:**

Revised as suggested.

**Changes in Manuscript:**

"…2% HNO$_3$…" and "…2% dilute nitric acid…".

4. Line 163: Besides heavy metal contents, were the mineral elements in dust aerosols detected?

**Response:**

This study primarily focuses on the enrichment factor of heavy metals, and we did not detect any mineral elements. This limitation arises from the detection range of our chosen method in ICPMS. Routine mineral elements, such as Na, K, S, Mg and Al, fall beyond the detection range of ICPMS.

5. Line 180:What are the types of dust weather? What type of dust weather was observed in 2018?

**Response:**

The "Dust Storm Weather Grade" is categorized into five levels based on the ground visibility during dust storm weather. These levels include floating dust, sand dust, sandstorm, strong sandstorm, and super strong sandstorm. On May 23$^{rd}$, 2018 (LT), the observed type of dust weather was floating dust.

**Changes in Manuscript:**

"On May 23$^{rd}$, 2018 (LT), on-site field measurements were conducted in Shanghai to assess the ambient dust particles. The measurements indicated an average wind speed of 2.2 m/s, which corresponds to a level of floating dust storm with a visibility of up to 10 km."

6. Line 193: "… CMAQ model v5.0.1 with an expanded SAPRC-99" Please show the full name for readers who are not familiar with CMAQ and SAPRC-99.

**Response:**

Corrected.

**Changes in Manuscript:**

"…Community Multiscale Air Quality (CMAQ) model v5.0.1 with an expanded Stratospheric and Air Pollution Research-99 (SAPRC-99) photochemical mechanism…".

7. Line 209: Please edit as "Inverse Distance Weight (IDW)".

**Response:**

Revised as suggested.

**Changes in Manuscript:**

"…Inverse Distance Weight (IDW) …".

8. Line 215: "E1, s1, and a are the values for dust." Please check the font format.

**Response:**

Corrected.

**Changes in Manuscript:**

"$E_1$, $s_1$, and $a$ are the values for dust."

9. Line 217-222 and Line 233-238: Please check line spacing as it should be consistent with context.

**Response:**

Changed as suggested.

We thank Referee 1 again for the comments and suggestions!

**Reference**

Alfaro, S. C.: Influence of soil texture on the binding energies of fine mineral dust particles potentially released by wind erosion, Geomorphology, 93, 157-167, 10.1016/j.geomorph.2007.02.012, 2008.

Alfaro, S. C., Gaudichet, A., Gomes, L., and Maille, M.: Modeling the size distribution of a soil aerosol produced by sandblasting, Journal of Geophysical Research-Atmospheres, 102, 11239-11249, 10.1029/97jd00403, 1997.

DoE, U.: The risk assessment information system (RAIS), Argonne, IL: US Department of Energy's Oak Ridge Operations Office (ORO), 2011.

Gao, Q., Zhu, X., Wang, Q., Zhou, K., Lu, X., Wang, Z., and Wang, X.: Enrichment and transfer of polycyclic aromatic hydrocarbons (PAHs) through dust aerosol generation from soil to the air, Frontiers of Environmental Science & Engineering, 17, 10, 2023.

Gholizadeh, A., Taghavi, M., Moslem, A., Neshat, A. A., Lari Najafi, M., Alahabadi, A., Ahmadi, E., Ebrahimi aval, H., Asour, A. A., Rezaei, H., Gholami, S., and Miri, M.: Ecological and health risk assessment of exposure to atmospheric heavy metals, Ecotoxicology and Environmental Safety, 184, 109622, https://doi.org/10.1016/j.ecoenv.2019.109622, 2019.

Kettler, T. A., Doran, J. W., and Gilbert, T. L.: Simplified Method for Soil Particle-Size Determination to Accompany Soil-Quality Analyses, Soil Science Society of America Journal, 65, 849-852, https://doi.org/10.2136/sssaj2001.653849x, 2001.

Kok, J. F.: Does the size distribution of mineral dust aerosols depend on the wind speed at emission?, Atmospheric Chemistry and Physics, 11, 10149-10156, 2011.

Lafon, S., Alfaro, S. C., Chevaillier, S., and Rajot, J. L.: A new generator for mineral dust aerosol production from soil samples in the laboratory: GAMEL, Aeolian Research, 15, 319-334, 10.1016/j.aeolia.2014.04.004, 2014.

Lafon, S., Sokolik, I. N., Rajot, J. L., Caquineau, S., and Gaudichet, A.: Characterization of iron oxides in mineral dust aerosols: Implications for light absorption, Journal of Geophysical Research-Atmospheres, 111, 10.1029/2005jd007016, 2006.

Luo, X.-S., Ding, J., Xu, B., Wang, Y.-J., Li, H.-B., and Yu, S.: Incorporating bioaccessibility into human health risk assessments of heavy metals in urban park soils, Science of The Total Environment, 424, 88-96, https://doi.org/10.1016/j.scitotenv.2012.02.053, 2012.

Moya, J., Phillips, L., Schuda, L., Wood, P., Diaz, A., Lee, R., Clickner, R., Birch, R., Adjei, N., and Blood, P.: Exposure factors handbook: 2011 edition, US Environmental Protection Agency, 2011.

---

## Author Comment (AC2)

**Responses to interactive comments**

**Journal: Atmospheric Chemistry and Physics**

**Manuscript ID: acp-2022-802**

**Title: High Enrichment of Heavy Metals in Fine Particulate Matter through Dust Aerosol Generation**

We appreciate Referee #2's comments and suggestions to help improve the manuscript. Every comment is addressed, and the detailed responses and related changes are shown below. Our response is in blue and the modifications in the manuscript are in red.

**General comments:**

The paper titled "High Enrichment of Heavy Metals in Fine Particulate Matter through Dust Aerosol Generation" by Gao et al. examined the enrichment of heavy metals in the laboratory-generated dust aerosols, which generated from soil samples that collected from dust source regions and typical cities. Then, by using a regional air quality model, the authors modeled the contribution of dust aerosol to atmospheric heavy metal loadings, based on dust aerosol profiles determined in present study as well as the SPECIATE profile from the US EPA's SPECIATE database, the comparison analysis suggested that usually using the SPECIATE profile in regional air quality models could not capture the correct size-dependent selectivity of heavy metals in dust aerosols, and would have significant errors in calculating contribution of fine dust aerosols to atmospheric heavy metals, as well as their cancer risks. The manuscript was well written and presented clearly. Therefore, I recommend the publication of Gao et al. work after some issues were properly revised and improved.

**Response:** We thank Referee 2 for the valuable comments and suggestions. Below are the responses to each specific comment and question.

**Specific and technical comments:**

1. Soil sample collection, it is not clear why the authors collected soil sample in Shanghai as it was not the dust source region. In addition, it is better to provide information on the dust events occurred in Shanghai and which dust source region influence Shanghai city most. Then, the motivation on the selection of soil sample would be more clear.

**Response:**

In this study, a total of 14 soil samples were analyzed, with 13 samples collected from dust source areas, such as S1-S13. The soil sample collected in Shanghai is considered as a local reference for comparison. During dry weather conditions, wind can also suspend dust aerosols from the soil surface in the Shanghai region, making it a significant local source of dust aerosols (Liu et al., 2016; Liu et al., 2020).

The north of China has been divided into four major dust storm source regions by the Chinese Ministry of Environmental Protection. These regions include the Gansu Hexi Corridor and Inner Mongolia's Alxa League, the surrounding areas of Taka Laka Mangal Desert in Xinjiang, the adjacent areas of Yin Shan North Slope and Hun Shan Dake Desert in Inner Mongolia, and the areas along the Great Wall of China near the boundaries of Mongolia and Ningxia. During the prevailing dust storm periods (March to May) in the East Asian region, there are significant increases in the concentrations of dust aerosols.

Dust events in Shanghai are primarily influenced by dust sources from the western Inner Mongolia Gobi, deserts in the Tibetan Plateau, and arid deserts in northwest China (Fu et al., 2010; Fu et al., 2014; Sun et al., 2017).

**Changes in Manuscript:**

"Although the soil (S14) collected in Shanghai does not originate from a dust source region, it can still produce dust aerosols in some cases. For example, under dry weather conditions, the soil surface in the Shanghai area could serve as a significant local contributor to the generation of dust aerosols (Liu et al., 2016; Liu et al., 2020). During the prevailing dust storm periods from March to May, Shanghai is primarily influenced by dust originating from the western Inner

Mongolia Gobi, deserts in the Tibetan Plateau, and arid deserts in northwest China (Fu et al., 2010; Fu et al., 2014; Sun et al., 2017)."

2. Line 208-210, what about the regions that between the dust sources?

**Response:**

In the manuscript, we have included the statement, "…outside these four regions were estimated using Inverse Distance Weight (IDW) spatial interpolation methods." Specifically, we utilized the Inverse Distance Weight (IDW) spatial interpolation methods to derive the emission factors (which refer to the amount of heavy metal emitted per kilogram of dust) for areas outside the four regions. This approach relied on the experimental dataset of emission factors within these four regions and is a commonly used method for estimating the spatial distribution of atmospheric pollutant variables (Zhang and Tripathi, 2018). The details of the IDW method could be found in the revised SI. Subsequently, we generated a map of emission factors for the four major dust regions and other regions in China. Furthermore, we employed the dataset of emission factors in the CMAQ model to simulate the spatial distributions of heavy metals originating from the dust sources both inside and outside the four major dust regions.

**Changes in Manuscript:**

"It is worth noting that the emission factors for areas outside these four regions were estimated using Inverse Distance Weight (IDW) spatial interpolation methods. These methods were based on the dataset of emission factors within these four regions, which represent the amount of heavy metal emitted per kilogram of dust (Zhang and Tripathi, 2018)."

**Texture S2.** Inverse Distance Weight (IDW)

IDW is a point based interpolation method (Harman et al., 2016). The value at point ($N_0$) is calculated through the following formula.

$$N_0 = \frac{\sum_{i=1}^{n} N_i \cdot P_i}{\sum_{i=1}^{n} P_{ii}} \qquad (1)$$

Where *n* represents the number of measurement points. $N_i$ represents the value at point *i*. $P_i$ is the weight of the value at *i* position. The weight $P_i$ can be calculated with Eq. (2) below as a function of the distance between the reference point and the interpolation point following from the idea that the effect of the closer points is higher than distance ones (Macedonio and Pareschi, 1991).

$$p_i = \frac{1}{d_i^k} \qquad i = 1, 2, \ldots n \tag{2}$$

Where $d_i$ is the horizontal distance between the interpolation point at ($x_0$, $y_0$) and the reference points at ($x_i$, $y_i$) and is calculated by Eq. (3). k is the power of the distance.

$$d_i = \sqrt{(x_i - x_0)^2 + (y_i - y_0)^2} \tag{3}$$

3. Line 259-260, the EFs value of 5 for Cd along has no meaning, could you provide some information on EFs from other studies and relate discussions here?

**Response:**

We have conducted a thorough review of relevant literature and included this discussion in the revised manuscript. No other literature has reported the enrichment of Cd or other heavy metals in dust aerosols. However, there is one study showing the enrichment of water-soluble ions during dust aerosol production from soil (Wu et al., 2022). It reports that the EFs of $Ca^{2+}$ ranged from approximately 5.6 to 223.1, and the EF values of $Mg^{2+}$ were between approximately 2.1 and 90.3 for dust-$PM_{2.5}$ from Sandy soils in the Taklamakan Desert. In this study, it is found that the EF of Cd and other metals falls within the range of EF for these water-soluble ions, consistent with the value reported by Wu et al., (2022).

**Changes in Manuscript:**

"No other literature has reported the enrichment of Cd or other heavy metals in dust aerosols. However, there is one study showing the enrichment of water-soluble ions during dust aerosol production from soil (Wu et al., 2022). It reports that the EFs of $Ca^{2+}$ ranged from approximately 5.6 to 223.1, and the EF values of $Mg^{2+}$ were between approximately 2.1 and 90.3 for dust-$PM_{2.5}$ from Sandy soils in the Taklamakan

Desert. In this study, it is found that the EF of Cd and other metals falls within the range of EF for these water-soluble ions, consistent with the value reported by Wu et al., (2022)."

1. Line 275-276, as six kinds of soil types (silty loam; sand; sandy loam; loam; loam sand; silty clay loam.) had been collected, I suggested the comparison analysis among different soil types or soil texture on heavy metals and their EFs needed to be conducted.

   **Response:**

   To examine the relationship between soil texture and their corresponding enrichment factors (EFs), we conducted a one-way Analysis of Variance (ANOVA) test using SPSS. ANOVA is a statistical method used to determine if there are any significant differences between the means of two or more groups. The *p-value* in ANOVA represents the probability of obtaining the observed differences in means (or more extreme differences) by random chance alone, assuming that there is no true difference between the groups. If the *p-value* is less than a predetermined significance level (commonly 0.05), it indicates that there are significant differences between the means of the groups being compared.

   In our study, we first compared the differences in EFs within the same soil texture. Specifically, for sandy soil, we found variations in the enrichment factors of heavy metal for dust-$PM_{2.5}$ (*p-value*=0.004<0.05) and dust-$PM_{10}$ (*p-value*=0<0.05). These results indicate that there are significant differences in the EFs of heavy metals within the sandy soil group.

   Furthermore, we compared the EFs among six different soil types. The ANOVA results indicated significant differences in the EFs of dust-$PM_{2.5}$ (*p-value*=0<0.05) and dust-$PM_{10}$ (*p-value*=0<0.05) among these soil types. The differences observed among the six different soil types were greater than those observed among the six sandy soils for dust-$PM_{2.5}$, suggesting a potential role of soil type in affecting EFs, which would require further study to elucidate.

**Changes in Manuscript:**

"When examining the impact of soil texture on dust aerosol enrichment, first, notable variations were observed in the EF values from one soil texture, such as sandy soils, specifically S2, S4, S7, S10, S11, and S12. To assess the significance of these variations, a one-way Analysis of Variance (ANOVA) was conducted using SPSS. In ANOVA, the *p-value* represents the probability of obtaining the observed differences in means (or more extreme differences) by random chance alone, assuming no true difference between the groups. A *p-value* below a predetermined significance level (commonly 0.05) indicates significant differences between the means of the compared groups. Specifically, for sandy soil, analysis results reveal significant variations between these six soils in terms of the EF values for both dust-$PM_{2.5}$ (*p-value*=0.004<0.05) and dust-$PM_{10}$ (*p-value*=0<0.05) (Table S5 and S6). These results indicate that there are significant differences in the EFs of heavy metals within the sandy soil group. Then, the variation between soil types was analyzed. For the six different types of soil samples, the results of ANOVA showed significant differences in the EFs of dust-$PM_{2.5}$ (*p-value*=0<0.05) and dust-$PM_{10}$ (*p-value* =0<0.05) among these soil types (Table S7 and S8). The differences among the six soils from different soil types were greater than those observed among the different soils in the same soil type, indicating a potential role of soil type in affecting EFs, which would require further study to elucidate. Detailed information was found in SI of Texture S3 and Table S5-S10."

**Text S3.** A one-way Analysis of Variance (ANOVA) analysis

To examine the relationship between soil texture and their corresponding enrichment factors (EFs), a one-way Analysis of Variance (ANOVA) test was conducted using SPSS. When comparing the differences among the six types of sandy soils (S2, S4, S7, S10, S11, and S12), enter the average EF values (dust-$PM_{2.5}$ and dust-$PM_{10}$) for the six types of sandy soils in the software, and then select one-way ANOVA with a confidence level of 0.05.

To compare the differences in enrichment factors among different soil types,

considering that the number of soil samples for each type was not equal, calculate the average enrichment factor for each type using two or more soil samples of the same type. Then, input the average enrichment factors (dust-$PM_{2.5}$ and dust-$PM_{10}$) for each type of soil (silty loam, sand, sandy loam, loam, loam sand, and silty clay loam) into the software and perform the aforementioned operations. The data and specific results can be found in Table S5-S8.

**Table S5.** A one-way Analysis of Variance (ANOVA) analysis was conducted for dust-$PM_{2.5}$ among sandy soils (S2, S4, S7, S10, S11, and S12).

| Origin of disparities | SS | df | MS | F | *P*-value | F crit |
|---|---|---|---|---|---|---|
| Between the group | 15.62294 | 5 | 3.124589 | 3.79773 | 0.004393 | 2.353809 |
| Within the group | 54.30161 | 66 | 0.822752 | | | |
| Total | 69.92456 | 71 | | | | |

**Table S6.** A one-way Analysis of Variance (ANOVA) analysis was conducted in dust-$PM_{10}$ among sandy soils (S2, S4, S7, S10, S11, and S12).

| Origin of disparities | SS | df | MS | F | *P*-value | F crit |
|---|---|---|---|---|---|---|
| Between the group | 14.74211 | 5 | 2.948422 | 31.17927 | 3.79E-16 | 2.353809 |
| Within the group | 6.241193 | 66 | 0.094564 | | | |
| Total | 20.9833 | 71 | | | | |

**Table S7.** A one-way Analysis of Variance (ANOVA) analysis was conducted in dust-$PM_{2.5}$ among six different soil types (silty loam; sand; sandy loam; loam; loam sand and silty clay loam).

| Origin of disparities | SS | df | MS | F | *P*-value | F crit |
|---|---|---|---|---|---|---|
| Between the group | 78.82538 | 5 | 15.76508 | 15.56416 | 4.28E-10 | 2.353809 |
| Within the group | 66.852 | 66 | 1.012909 | | | |
| Total | 145.6774 | 71 | | | | |

**Table S8.** A one-way Analysis of Variance (ANOVA) analysis was conducted in dust-$PM_{10}$ among six different soil types (silty loam; sand; sandy loam; loam; loam sand and silty clay loam).

| Origin of disparities | SS | df | MS | F | $P$-value | F crit |
|---|---|---|---|---|---|---|
| Between the group | 6.130101 | 5 | 1.22602 | 19.79507 | 5.35E-12 | 2.353809 |
| Within the group | 4.087752 | 66 | 0.061936 | | | |
| | | | | | | |
| Total | 10.21785 | 71 | | | | |

2. Line 282-283, I note that only one soil sample (S10) was chosen to explore its particle size distribution and associated EFs. Did the authors also investigate the other soil samples? And why?

    **Response:**

    We focused our investigation solely on the S10 soil sample to examine its particle size distribution and associated EFs. Here are the reasons:

    First, S10 is sampled from the western Inner Mongolia Gobi, which serves as a representative dust source area that impacts the Shanghai region during dust storm events. Thus, S10 can serve as a representative soil sample for our study.

    Second, the MOUDI experiment is a very labor-intensive process that requires at least three replicates, each involving seven PVC filters capturing particles with different size ranges. One set of experiments would produce at least 21 filters that need to be analyzed with the offline techniques. Conducting MOUDI experiments with multiple soils would require significantly more effort and cost.

    Therefore, here we only used the S10 soil sample for the MOUDI experiment.

3. Line 285-290, the discussion on the dust particle size distribution is limited. More information on the heavy metals presented in Fig,2 should be provided. I note that the EFs of some heavy metals increased with decreased particle size, some showed no changes and one heavy metal (Ti) showed reverse variations. These interesting results should be provided.

    **Response:**

We have some additional discussion regarding the correlation between dust particle size distribution and EFs for various heavy metals, including V, Cr, Co, Mn, Ni, Cu, Zn, As, and Ba. It was observed that the EFs of these metals increase as the particle size decreases. However, the EFs for Cd do not show any significant variation with particle size. Interestingly, the EFs of Ti exhibit an opposite trend, increasing as the particle size increases. We added some more discussion on these findings below.

**Changes in Manuscript:**

"V, Cr, Co, Mn, Ni, Cu, Zn, As, and Ba show consistent trends, with EFs increasing as the particle size decreases. In detail, V (ranging from ~1.1 to ~18.9), Cr (ranging from ~1.5 to ~23.7), Co (ranging from ~1.7 to ~93.7), Mn (ranging from ~2.3 to ~7.4), Ni (ranging from ~1.6 to ~29.7), Cu (ranging from ~3.3 to ~54.3), Zn (ranging from ~2.3 to ~19.0), As (ranging from ~1.8 to ~112.3), and Ba (ranging from ~1.4 to ~7.0), as the particle size decreases from 10 μm to 0.56 μm.

In contrast, Cd's EFs remain relatively unchanged with varying particle sizes. On the other hand, Ti exhibits an opposite trend, with EF values decreasing as the particle size decreasing, and the reason for this difference requires further study."

4. Line 315-323, for the modelled heavy metals concentrations, it is between to include the comparison discussion with the field observation results as plenty of particle chemical composition data in dust source regions and megacities had been published. Then, the author could evaluate the errors of using SPECIATE profile and the improvement in dust profiles conducted in present study.

**Response:**

We have thoroughly considered the comparison between our findings and field observations, as well as the evaluation of errors for specific elements in both dust source regions and megacities. However, it is important to note that our model represents the annual average data for the year 2013. Despite the existence of field studies conducted in the same year (Wang et al., 2021; Shi et al., 2018), they provide

additional insights.

In Wang et al.'s study, atmospheric heavy metal pollution in different regions of China over the past 30 years were analyzed and summarized. The analysis focused on the regional pollution characteristics of seven heavy metal elements, including As, Zn, Cr, Pb, Cd, Mn, and Ni in $PM_{2.5}$. The study revealed that regions with high levels of heavy metals in $PM_{2.5}$ were mainly concentrated in economically developed areas such as North China, East China, and South China. For example, in Baoding, a city in the North China region, the concentration of Pb was found to be 192.30 $ng/m^3$ in 2013, possibly attributed to metal smelting.

In Shi et al.'s research, $PM_{2.5}$ samples were collected in April and October 2013 in Kunming city. The study investigated Cr, Mn, Pb, Ni, Cu, Zn, As, and Cd. The results indicated that the mass concentrations of Mn, Pb, Ni, Cu, Zn, As, and Cd in $PM_{2.5}$ were higher in the industrial area monitoring site compared to the traffic-intensive area and the clean control site. Additionally, heavy metal concentrations were generally higher in winter and spring compared to summer and autumn. Principal component analysis suggested that the main sources of heavy metals in $PM_{2.5}$ in the urban area of Kunming were metallurgical industries (49.43%), a mixture of dust from the ground and road traffic (18.73%), and coal combustion (12.61%).

As mentioned above, while these studies provide valuable insights, we were unable to obtain annual average data for a direct comparison with our model results.

**Changes in Manuscript:**

"Uncertainties associated with the use of SPECIATE have also been identified in previous studies (Ho et al., 2003; Xia et al., 2017). Specifically, the dust $PM_{2.5}$ source profiles obtained from local studies indicated that SPECIATE overestimated the contributions of atmospheric K and Al by approximately 23%, while underestimating the contributions of Ca and Na by 50%. Additionally, the model represents the annual average data for the year 2013. Although there are some field studies conducted in the same year (Wang et al., 2021; Shi et al., 2018), there is no

available annual average data for a direct comparison with the model results."

5. Line 313, it should be Cu not Cr that present in Fig S12.

   **Response:**

   Thanks for your comment. Revised accordingly.

   **Changes in Manuscript:**

   "…Cu…".

6. Line 331-332, why the simulated areas were different by applying different profiles?

   **Response:**

   Thanks for pointing this out. The writing here is not clear and confusing. Some previous studies (Gunawardana et al., 2012; Zhuang et al., 2001) have made an assumption that the composition of dust aerosols is similar to that of its parent soil. Thus, we apply the profiles from soil composition and dust-$PM_{2.5}$ to the model and investigate the difference, which is indeed evident. For example, when applying dust-$PM_{2.5}$ profiles, the contribution of dust aerosols to atmospheric Cr ranged from 0.02 to 0.08 $\mu g/m^3$ over a larger geographical area in China. Whereas using soil profiles, it was observed that dust aerosols contributed to atmospheric Cr levels ranging from 0.02 to 0.08 $\mu g/m^3$ within a much smaller geographical area in China. We have revised the sentence to avoid any potential misunderstanding.

   **Changes in Manuscript:**

   "In contrast, the application of the soil profile to the model reveals a significantly reduced area where the modeled Cr concentration from dust aerosols falls within the range of 0.02 to 0.08 $\mu g/m^3$."

7. Line 368-369, before comparison with the field observations of ambient PM2.5, it could not be concluded that dust aerosol could be the main sources of multiple heavy metals in China.

   **Response:**

   Thanks for this comment. This sentence was modified and shown below.

**Changes in Manuscript:**

"Our modeling results suggest that dust aerosol could be a major source of multiple heavy metals in PM$_{2.5}$ in China."

Again, we thank the Referee for all the valuable questions and suggestions, which have helped improve our work greatly!

**Reference**

Fu, Q., Zhuang, G., Li, J., Huang, K., Wang, Q., Zhang, R., Fu, J., Lu, T., Chen, M., Wang, Q., Chen, Y., Xu, C., and Hou, B.: Source, long-range transport, and characteristics of a heavy dust pollution event in Shanghai, Journal of Geophysical Research: Atmospheres, 115, https://doi.org/10.1029/2009JD013208, 2010.

Fu, X., Wang, S. X., Cheng, Z., Xing, J., Zhao, B., Wang, J. D., and Hao, J. M.: Source, transport and impacts of a heavy dust event in the Yangtze River Delta, China, in 2011, Atmospheric Chemistry and Physics, 14, 1239-1254, 10.5194/acp-14-1239-2014, 2014.

Gunawardana, C., Goonetilleke, A., Egodawatta, P., Dawes, L., and Kokot, S.: Source characterisation of road dust based on chemical and mineralogical composition, Chemosphere, 87, 163-170, 10.1016/j.chemosphere.2011.12.012, 2012.

Harman, B. I., Koseoglu, H., and Yigit, C. O.: Performance evaluation of IDW, Kriging and multiquadric interpolation methods in producing noise mapping: A case study at the city of Isparta, Turkey, Applied Acoustics, 112, 147-157, 10.1016/j.apacoust.2016.05.024, 2016.

Ho, K. F., Lee, S. C., Chow, J. C., and Watson, J. G.: Characterization of PM10 and PM2.5 source profiles for fugitive dust in Hong Kong, Atmospheric Environment, 37, 1023-1032, 10.1016/s1352-2310(02)01028-2, 2003.

Liu, Q., Liu, X., Liu, T., Kang, Y., Chen, Y., Li, J., and Zhang, H.: Seasonal variation in particle contribution and aerosol types in Shanghai based on satellite data from MODIS and CALIOP, Particuology, 51, 18-25, https://doi.org/10.1016/j.partic.2019.10.001, 2020.

Liu, Q., Wang, Y., Kuang, Z., Fang, S., Chen, Y., Kang, Y., Zhang, H., Wang, D., and Fu, Y.: Vertical distributions of aerosol optical properties during haze and floating dust weather in Shanghai, Journal of Meteorological Research, 30, 598-613, 10.1007/s13351-016-5092-4, 2016.

Macedonio, G. and Pareschi, M. T.: An algorithm for the triangulation of arbitrarily distributed points - Applications to volume estimate and terrain fitting Computers & Geosciences, 17, 859-874, 10.1016/0098-3004(91)90086-s, 1991.

Shi, J., Li, Z., Sun, Z., Han, X., Shi, Z., Xiang, F., and Ning, P.: Specific features of heavy metal pollutant residue in PM2. 5 and analysis of their damage level for human health in the urban air of Kunming, J. Saf. Environ, 18, 795-800, 2018.

Sun, R., Wang, H., Ma, X., Chen, Y., Zhao, B., Qin, Y., Zhang, H., and Ye, W.: Aerosol optical properties and formation mechanism of a typical air pollution episode in Shanghai during different weather condition periods, Acta Scientiae Circumstantiae, 37, 814-823, 2017.

Wang, L., Li, H., Zhang, W., Qi, J., Tian, H., Huang, K., Chen, D., and Guo, J.: Regional Pollution Characteristics of Heavy Metals in PM2.5, Research of Environmental Sciences, 34, 849-862, 2021.

Wu, F., Cheng, Y., Hu, T., Song, N., Zhang, F., Shi, Z., Hang Ho, S. S., Cao, J., and Zhang, D.: Saltation–Sandblasting Processes Driving Enrichment of Water-Soluble Salts in Mineral Dust, Environmental Science & Technology Letters, 2022.

Xia, Z., Fan, X., Huang, Z., Liu, Y., Yin, X., Ye, X., and Zheng, J.: Comparison of Domestic and Foreign PM_(2.5)Source Profiles and Influence on Air Quality Simulation, Research of Environmental Sciences, 30, 359-367, 2017.

Zhang, H. R. and Tripathi, N. K.: Geospatial hot spot analysis of lung cancer patients correlated to fine particulate matter (PM2.5) and industrial wind in Eastern Thailand, Journal of Cleaner Production, 170, 407-424, 10.1016/j.jclepro.2017.09.185, 2018.

Zhuang, G. S., Guo, J. H., Yuan, H., and Zhao, C. Y.: The compositions, sources, and size distribution of the dust storm from China in spring of 2000 and its impact on the global environment, Chinese Science Bulletin, 46, 895-901, 10.1007/bf02900460, 2001.

---

## Author Response (AR2)

**Responses to interactive comments**

**Journal: Atmospheric Chemistry and Physics**

**Manuscript ID: acp-2022-802**

**Title: High Enrichment of Heavy Metals in Fine Particulate Matter through Dust Aerosol Generation**

We appreciate Referee #1's comments and suggestions to help improve the manuscript. Our response is in blue and the modifications in the manuscript are in red.

| | |
|---|---|
| **Anonymous during peer-review:** | **Yes** No |
| **Anonymous in acknowledgements of published article: Yes** No | |

**Checklist for reviewers**

| | |
|---|---|
| **1) Scientific significance**

Does the manuscript represent a substantial contribution to scientific progress within the scope of this journal (substantial new concepts, ideas, methods, or data)? | Outstanding **Excellent** Good Fair Low |
| **2) Scientific quality**

Are the scientific approach and applied methods valid? Are the results discussed in an appropriate and balanced way (consideration of related work, including appropriate references)? | Outstanding **Excellent** Good Fair Low |
| **3) Presentation quality**

Are the scientific results and conclusions presented in a clear, concise, and well | **Outstanding** Excellent Good Fair Low |

| structured way (number and quality of figures/tables, appropriate use of English language)? | |
|---|---|

**For final publication, the manuscript should be**

**accepted as is**

accepted subject to **technical corrections**

accepted subject to **minor revisions**

reconsidered after **major revisions**

**rejected**

**Were a revised manuscript to be sent for another round of reviews:**

I would be willing to review the revised manuscript.

**I would not be willing to review the revised manuscript.**

**Suggestions for revision or reasons for rejection**

(visible to the public if the article is accepted and published)

Author addressed all my comments, and I suggest to accept the revised version.

**Response:**

We thank Referee 1 again for the comments and suggestions and found that Referee #1 does not have any more comments that need to be revised.

**Responses to interactive comments**

**Journal: Atmospheric Chemistry and Physics**

**Manuscript ID: acp-2022-802**

**Title: High Enrichment of Heavy Metals in Fine Particulate Matter through Dust Aerosol Generation**

We appreciate Referee #2's comments and suggestions to help improve the manuscript. Our response is in blue and the modifications in the manuscript are in red.

**Anonymous during peer-review:**          **Yes** No

**Anonymous in acknowledgements of published article: Yes** No

**Checklist for reviewers**

| | |
|---|---|
| **1) Scientific significance**
Does the manuscript represent a substantial contribution to scientific progress within the scope of this journal (substantial new concepts, ideas, methods, or data)? | Outstanding Excellent **Good** Fair Low |
| **2) Scientific quality**
Are the scientific approach and applied methods valid? Are the results discussed in an appropriate and balanced way (consideration of related work, including appropriate references)? | Outstanding **Excellent** Good Fair Low |
| **3) Presentation quality**
Are the scientific results and conclusions presented in a clear, concise, and well | Outstanding **Excellent** Good Fair Low |

| structured way (number and quality of figures/tables, appropriate use of English language)? | |
|---|---|

**For final publication, the manuscript should be**

**accepted as is**

accepted subject to **technical corrections**

accepted subject to **minor revisions**

reconsidered after **major revisions**

**rejected**

**Were a revised manuscript to be sent for another round of reviews:**

I would be willing to review the revised manuscript.

**I would not be willing to review the revised manuscript.**

**Suggestions for revision or reasons for rejection**

(visible to the public if the article is accepted and published)
* * *
**Response:**

Again, we thank the Referee 2 for all the valuable questions and suggestions and found that Referee #2 does not have any more comments that need to be revised.

Furthermore, we have reviewed the entire manuscript and made some minor corrections.

**Changes in manuscript:**

**Line 159:** "…a 10-stage…"

**Line 178:** "…an…"

**Line 197:** "…a floating dust storm…"

**Line 200 and Line 432:** "…single-particle…"

**Line 239:** "…the skin…"

**Line 337:** "These results…"

**Line 394:** "…a value…"

**Line 488:** "…compared to…"

**Line 493:** "…which seems to lack enrichment…"